# On an Integral Equation with the Riemann Function Kernel

**Sergei Sitnik * and Abdul Ahad Arian**

Chair of Applied Mathematics and Computer Modelling, Institute of Engineering and Digital Technologies, Belgorod State National Research University, Pobedy Street, 85, 308015 Belgorod, Russia; arinkandahar@mail.ru
* Correspondence: sitnik@bsu.edu.ru

**Abstract:** This paper is concerned with a study of a special integral equation. This integral equation arises in many applied problems, including transmutation theory, inverse scattering problems, the solution of singular Sturm–Liouville and Shrödinger equations, and the representation of solutions of singular Sturm–Liouville and Shrödinger equations. A special integral equation is derived and formulated using the Riemann function of a singular hyperbolic equation. In the paper, the existence of a unique solution to this equation is proven by the method of successive approximations. The results can be applied, for example, to representations of solutions to Sturm–Liouville equations with singular potentials, such as Bargmann and Miura potentials, and similiar. The treatment of problems with such potentials are very important in mathematical physics, and inverse, scattering and related problems. The estimates received do not contain any undefined constants, and for transmutation kernels all estimates are explicitly written.

**Keywords:** transmutations; Sturm–Liouville operator; singular potential; Bargmann potential; successive approximations

**MSC:** 34B24; 34L25

## 1. Introduction

In this section, we introduce the problem of the integral representation of solutions to Sturm–Liouville equations with singular potential by a transmutation operator of the Poisson type.

As, in fact, we use an underlying idea of transmutations in this paper, let us give a proper definition.

**Definition 1.** *For a given pair of operators $(A, B)$ and an operator $T$, $T \neq 0$ is called the transmutation (or intertwining) operator if, on elements of some functional spaces, the next property is valid*

$$T A = B T. \tag{1}$$

It is obvious that the notion of transmutation is direct and far reaching generalization of the similarity notion from linear algebra. However, transmutations do not reduce to similar operators because intertwining operators often are not bounded in classical spaces and the inverse operator may not exist or be bound in the same space. As a consequence, spectra of transmuted operators $A$, $B$ are not the same, as a rule, in contrast to to a case of similar operators. Moreover, transmutations may be unbound. Additionally, a pair of intertwining operators may not be differential ones. In transmutation theory, there are problems for next varied types of operators: integral, integro–differential, difference–differential (e.g., the Dunkl operator), pseudodifferential and abstract differential operators, cf. [1–8]. In quantum physics, in the study of the Shrödinger equation and inverse scattering theory, underlying transmutations are called wave operators.

Additionally, how do transmutations usually work? Suppose we study properties for a rather complicated operator $A$. However, suppose also that we know corresponding properties for the model, which has a more simple operator $B$ and transmutation $T$; in (1) this readily exists. Then, we may usually translate results for the model operator $B$ to a more complicated operator $A$. This is, briefly, the main idea of transmutations.

Let us, for example, consider an equation $Au = f$, then, applying to it a transmutation with property (1), we consider a new equation: $Bv = g$, with $v = Tu, g = Tf$. Therefore, if we can solve the simpler equation $Bv = g$ then the initial one is also solved and has solution $u = T^{-1}v$. Of course, it is supposed that the inverse operator exists and its explicit form is known. This is a simple application of the transmutation technique for proving formulas for the solutions of ordinary and partial differential equations, cf. [9] for more details.

For the explicit construction of transmutations, a special method was introduced and developed by the first author—the *integral transforms composition method (ITCM)*, thoroughly studied in [3,4,9] (and more references therein). The essence of this method is to construct the necessary transmutation operator and corresponding connection formulas among the solutions of perturbed and nonperturbed equations, as a composition of classical integral transforms with properly chosen weighted functions.

In this paper, we consider the above described transmutation technique for finding a solution of a perturbed differential equation with a singular coefficient via solutions of an unperturbed differential equation, namely, via the Bessel functions.

Therefore, we consider the next problem in detail. Find a solution to the differential equation

$$B_\alpha u(x) - q(x)u(x) = u''(x) + \frac{2\alpha}{x}u'(x) - q(x)u(x) = 0, \ x > 0, \ \alpha > 0. \tag{2}$$

in the integral form

$$u(x) = P_\alpha v(x) = v(x) + \int_x^\infty P(x,t)v(t)\, dt, \tag{3}$$

where $v(x)$ is the Bessel function and $P(x,t)$ is a kernel function.

$$v(x) = J_\alpha(x),$$

and $P(x,t)$ is a kernel function.

In fact, (3) is a transmutation operator due to definition (1), with the choice

$$A = B_\alpha - q(x), \ \ B = B_\alpha, \ \ B_\alpha u(x) = u''(x) + \frac{2\alpha}{x}u'(x), \ \ T = P_\alpha.$$

Such a transmutation operator $P_\alpha$ (3), by the terminology of transmutations connected with the Bessel differential operator, is called a transmutation of the Poisson type for Equation (2). This connects solutions of Equation (2), including the potential function $q(x)$, with a more simple equation than the form of (2), but with $q(x) = 0$, namely, with the Bessel functions.

As a result, the transmutation operator (3) acts by the formula

$$P_\alpha(B_\alpha - q(x))v = B_\alpha P_\alpha v,$$

on proper functions.

This approach produces connection formulas between different solutions to differential equations, namely, of perturbed more complex ones and more simple model ones.

Transmutation theory is a rich and important field of modern mathematics. It has many applications in all areas of theoretical and applied mathematics, cf. [1–8]. The solution representation of the form (3) with a "good" kernel $P$ for a large variety of potentials $q(x)$ is basic for classical methods of solving direct and inverse problems, including inverse

scattering problems [1,4–6,10]. For Sturm–Liouville operators, transmutations and integral representation (3) were first introduced by B.Ya. Levin, cf. [1,2,4,7,8].

After that, in a series of papers, transmutations and inverse problems were also considered for perturbed Bessel operators with potentials, cf. [11–13]. Along with Poisson operators, their inverse Sonine transmutation operators $\mathbf{S}_\alpha$ were studied, which satisfy

$$S_\alpha(B_\alpha - q(x))u = B_\alpha S_\alpha u$$

on proper functions. These names are originated from formulas in special function theory [14]. Sonine and Poisson transmutations for the Bessel operator were introduced by J.Delsart, cf. [1,2,4,15]. The original method for constructing transmutations for the perturbed Bessel equation on a half-axis was developed by V.V. Stashevskaya in [11]. The method works for singular at zero potentials with an estimate $|q(x)| \leq cx^{-3/2+\varepsilon}$, $\varepsilon > 0$ for integer $\alpha$, this approach later received broad generalizability. A case of a continuous potential $q$, $\alpha > 0$ was studied in detail by A.S.Sokhin in [12,13], and, after that, by other authors.

In many problems in mathematics and physics, we have to study strongly singular potentials, e.g., with an arbitrary singularity at zero. In this paper, the problem of integral representation for similar problems is considered. In addition, we concentrate on solving a connected integral equation that produces such representations. In particular, such singular potentials are included as the singular $q = x^{-2}$, a strongly singular potential with power singularity $q = x^{-2-\varepsilon}$, $\varepsilon > 0$, Yukawa, Bargmann and Bateman–Shadan potentials [10]. Such versality is an advantage of methods based on Levin type transmutations.

In the paper, we study, as a main object, an integral equation for the kernel of the transmutation in need. After reducing the problem to an integral equation, we prove the existence and uniqueness of its solutions, and the existence of its necessary derivatives. In addition, sharp estimates are proven for the solution via a parameter and the potential of the Equation (2), via the special Legendre functions. For the special case of power potentials, the estimates are simplified. We use a technique based on using the Riemann function for Euler–Poisson–Darboux equations, estimating integrals by the Mellin transform technique and Slater–Marichev theorem.

Let us note that, in this paper, a special class of transmutations is introduced, which differs from known ones by some details. Usually, the same limits are considered in integral equations, both as $[0; a]$ **OR** $[a; \infty]$ in the main integral equations for the kernel function of a transmutation operator. In this paper, we use two different limits: $[0; a]$ **AND** $[a; \infty]$. This approach leads to a wider class of permitted singular potentials.

Furthermore, we modernize a little the method of Darboux–Levitan from [15], on exploiting the basic integral equation. It turned out that the Riemann function in this equation is represented not only by the Gauss hypergeometric function, but by the Legendre function. This simplifies estimates, as a special function with three parameters is reduced to a more simple special function with two parameters.

## 2. Derivation and Solution of an Integral Equation for the Transmutation Kernel

Let us introduce new variables and functions with the formulas

$$\xi = \frac{t + x}{2}, \ \eta = \frac{t - x}{2}, \ \xi \geq \eta > 0;$$

$$K(x, t) = \left(\frac{x}{t}\right)^\alpha P(x, t), \ u(\xi, \eta) = K(\xi - \eta, \xi + \eta). \tag{4}$$

Let $\nu = \alpha - 1$. Therefore, to derive representation (3) for Equation (2), it is enough to find function $u(\xi, \eta)$. It is well known (cf., [15]) that, if twice a differentiable solution $u(\xi, \eta)$ exists for an integral equation

$$u(\xi, \eta) = -\frac{1}{2} \int_{\xi}^{\infty} R_{\nu}(s, 0; \xi, \eta) q(s) \, ds - \int_{\xi}^{\infty} ds \int_{0}^{\eta} q(s + \tau) R_{\nu}(s, \tau; \xi, \eta) u(s, \tau) \, d\tau,$$

under conditions $0 < \tau < \eta < \xi < s$, then the function $P(x, t)$ is defined by (4), via this solution $u(\xi, \eta)$. The function $R_{\nu} = R_{\alpha-1}$ is the Riemann function appearing for the next singular inhomogeneous hyperbolic equation of the Euler–Poisson–Darboux kind

$$\frac{\partial^2 u(\xi, \eta)}{\partial \xi \partial \eta} + \frac{4\alpha(\alpha - 1)\xi\eta}{(\xi^2 - \eta^2)^2} u(\xi, \eta) = f(\xi, \eta),$$

which, in our case, is reduced to

$$\frac{\partial^2 u(\xi, \eta)}{\partial \xi \partial \eta} + \frac{4\alpha(\alpha - 1)\xi\eta}{(\xi^2 - \eta^2)^2} u(\xi, \eta) = q(\xi + \eta) u(\xi, \eta).$$

This Riemann function is known in the explicit form, cf. [15], via the Gauss hypergeometric function $_2F_1$:

$$R_{\nu} = \left( \frac{s^2 - \eta^2}{s^2 - \tau^2} \cdot \frac{\xi^2 - \tau^2}{\xi^2 - \eta^2} \right)^{\nu} {}_2F_1 \left( -\nu, -\nu; 1; \frac{s^2 - \xi^2}{s^2 - \eta^2} \cdot \frac{\eta^2 - \tau^2}{\xi^2 - \tau^2} \right). \tag{5}$$

This function was simplified in [? ], where it was expressed via the Legendre function

$$R_{\nu}(s, \tau, \xi, \eta) = P_{\nu} \left( \frac{1 + A}{1 - A} \right), \quad A = \frac{\eta^2 - \tau^2}{\xi^2 - \tau^2} \cdot \frac{s^2 - \xi^2}{s^2 - \eta^2} . \tag{6}$$

The main result of this paper is the next theorem.

**Theorem 1.** *Let a function $q(r) \in C^1(0, \infty)$ satisfy*

$$|q(s + \tau)| \le |p(s)|, \ \forall s, \forall \tau, \ 0 < \tau < s, \ \int_{\xi}^{\infty} |p(t)| \, dt < \infty, \forall \xi > 0. \tag{7}$$

*Then there exists an integral representation (3), in which the kernel function satisfies an estimate*

$$|P(r, t)| \le \left( \frac{t}{r} \right)^{\alpha} \frac{1}{2} \int_{\frac{t+r}{2}}^{\infty} P_{\alpha-1} \left( \frac{y^2(t^2 + r^2) - (t^2 - r^2)}{2try^2} \right) |p(y)| \, dy \ \cdot \tag{8}$$

$$\cdot \exp \left[ \left( \frac{t - r}{2} \right) \frac{1}{2} \int_{\frac{t+r}{2}}^{\infty} P_{\alpha-1} \left( \frac{y^2(t^2 + r^2) - (t^2 - r^2)}{2try^2} \right) |p(y)| \, dy \right].$$

*Further, the transmutation kernel $P(x, t)$ and also a solution to Equation (2) are twice continuously differentiable functions, according to both of their arguments in corresponding sets of definitions.*

We divide the proof of Theorem 1 to some lemmas.

Let us denote

$$I_q(\xi, \eta) = \frac{1}{2} \int\limits_{\xi}^{\infty} R_\nu(y, 0; \xi, \eta) |p(y)| \, dy = \frac{1}{2} \int\limits_{\xi}^{\infty} P_\nu \left( \frac{y^2(\xi^2 + \eta^2) - 2\xi^2\eta^2}{y^2(\xi^2 - \eta^2)} \right) |p(y)| \, dy, \quad (9)$$

$$u_0(\xi, \eta) = -\frac{1}{2} \int\limits_{\xi}^{\infty} R_\nu(s, 0; \xi, \eta) |p(s)| \, ds, \tag{10}$$

$$\mathbf{A}u_0(\xi, \eta) = - \int\limits_{\xi}^{\infty} ds \int\limits_{0}^{\eta} q(s + \tau) R_\nu(s, \tau; \xi, \eta) u_0(s, \tau) \, d\tau.$$

Prove the uniform convergence of von Neumann series

$$\sum_{k=0}^{\infty} \mathbf{A}^k u_0(\xi, y) \tag{11}$$

and note that it is twice differentiable.

**Lemma 1.** *The next estimate holds*

$$|u_0(\xi, \eta)| \leq I_q(\xi, \eta). \tag{12}$$

The proof is immediate from definitions (9) and (10).

**Lemma 2.** *Let $0 < \tau < \eta < \xi < s$. Then the next inequality holds*

$$I_q(s, t) \leq I_q(\xi, \eta). \tag{13}$$

**Proof.** We have $0 < \tau < \eta < \xi < s < y$. Let us derive, then

$$\frac{\tau^2}{s^2} \cdot \frac{(y^2 - s^2)}{(y^2 - \tau^2)} \leq \frac{\eta^2}{\xi^2} \cdot \frac{(y^2 - \xi^2)}{(y^2 - \eta^2)} \ (\leq 1).$$

In addition, this inequality is genuinely equivalent to

$$\tau^2 \xi^2 (y^2 - s^2)(y^2 - \eta^2) \leq \eta^2 s^2 (y^2 - \xi^2)(y^2 - \tau^2),$$

which is obvious, due to the fact that every multiplier from the left side is less than or equal to the corresponding multiplier from the right side. Further, consider, for $0 < x < 1$, a function

$$f(x) = \frac{1 + x}{1 - x} \geq 1, \ f'(x) = \frac{2}{(1 - x)^2} > 0, \ 0 < x < 1.$$

Consequently, this function is increasing in $x$, so

$$\frac{1 + \frac{\tau^2}{s^2} \cdot \frac{(y^2 - s^2)}{(y^2 - \tau^2)}}{1 - \frac{\tau^2}{s^2} \cdot \frac{(y^2 - s^2)}{(y^2 - \tau^2)}} \leq \frac{1 + \frac{\eta^2}{\xi^2} \cdot \frac{(y^2 - \xi^2)}{(y^2 - \eta^2)}}{1 - \frac{\eta^2}{\xi^2} \cdot \frac{(y^2 - \xi^2)}{(y^2 - \eta^2)}}.$$

The Legendre function $P_\nu(x)$ on $x \in (1, \infty)$ for $\nu > -1$ is increasing, and even more for $P_\nu(x) > 1$. It follows

$$P_\nu \left( \frac{1 + \frac{\tau^2}{s^2} \cdot \frac{(y^2 - s^2)}{(y^2 - \tau^2)}}{1 - \frac{\tau^2}{s^2} \cdot \frac{(y^2 - s^2)}{(y^2 - \tau^2)}} \right) \leq P_\nu \left( \frac{1 + \frac{\eta^2}{\xi^2} \cdot \frac{(y^2 - \xi^2)}{(y^2 - \eta^2)}}{1 - \frac{\eta^2}{\xi^2} \cdot \frac{(y^2 - \xi^2)}{(y^2 - \eta^2)}} \right).$$

The next inequality may be rewritten as

$$P_\nu\left(\frac{y^2(s^2+\tau^2)-2s^2\tau^2}{y^2(s^2-\tau^2)}\right) \le P_\nu\left(\frac{y^2(\xi^2+\eta^2)-2\xi^2\eta^2}{y^2(\xi^2-\eta^2)}\right).$$

Note that, in fact, we proved an inequality for the Riemann function

$$R_\nu(y,0;s,\tau) \le R_\nu(y,0;\xi,\eta) \tag{14}$$

for $0 < \tau < \eta < \xi < s < y$.

>From the above, the next estimate follows

$$I_q(s,\tau) = \frac{1}{2}\int\limits_s^\infty R_\nu(y,0;s,\tau)|p(y)|\,dy \le \frac{1}{2}\int\limits_\xi^\infty R_\nu(y,0;s,\tau)|p(y)|\,dy.$$

Changing the lower limit of integration $s$ to $\xi < s$, we may only increase the integral value, because the Riemann function is positive, $R_\nu > 0$. This leads to the desired estimate (13), so the Lemma 2 is proven. □

**Lemma 3.** *For the nth component of von Neumann series* (11), *the next estimate holds*

$$|u_n(\xi,\eta)| \le I_q(\xi,\eta)\cdot\frac{[\eta I_q(\xi,\eta)]^n}{n!}. \tag{15}$$

**Proof.** Let us use an inductive method. For $n = 0$, an inequality (15) is reduced to the result of Lemma 2. Let (15) be fulfilled for $n = k$. Then, for the next series component, it follows

$$|u_{k+1}(\xi,\eta)| \le \left|\int\limits_\xi^\infty ds \int\limits_0^\eta R_\nu(s,\tau;\xi,\eta)u_k(s,\tau)q(s+\tau)\,d\tau\right| \le$$

$$\le \int\limits_\xi^\infty ds \int\limits_0^\eta R_\nu(s,\tau;\xi,\eta)|q(s+\tau)|I_q(s,\tau)\frac{[\eta I_q(s,\tau)]^k}{k!}\,d\tau.$$

Repeating the arguments of the preceding lemma, we derive that

$$R_\nu(s,\tau;\xi,\eta) \le R_\nu(s,0;\xi,\eta), \tag{16}$$

due to

$$R_\nu(s,\tau;\xi,\eta) = P_\nu\left(\frac{1+A}{1-A}\right),\ A = \frac{\eta^2-\tau^2}{\xi^2-\tau^2}\cdot\frac{s^2-\xi^2}{s^2-\eta^2},$$

and the maximal $\tau$ value of $A$ is for $\tau = 0$. Now, using the inequalities (16) and (15), we derive

$$|u_{k+1}(\xi,\eta)| \le I_q(\xi,\eta)\frac{[\tau I_q(\xi,\eta)]^k}{k!}\cdot\int\limits_\xi^\infty R_\nu(s,0;\xi,\eta)\int\limits_0^\eta |q(s+\tau)|\tau^k\,d\tau\,ds.$$

We consider potentials obeying an inequality $|q(s+\tau)| \le |p(s)|$, $0 < \tau < s$. Finally, it follows

$$|u_{k+1}(\xi,\eta)| \le I_q(\xi,\eta)\frac{[I_q(\xi,\eta)]^{k+1}}{k!}\cdot\frac{\eta^{k+1}}{(k+1)},$$

and that completely proves an estimate (15) for all $n$. This proves the Lemma 3. □

Now we are ready to complete the proof of Theorem 1. Summing up all estimates (15) for all $n$, we derive that the von Neumann series is uniformly convergent in the variable domain $0 < \eta < \xi$ and its sum is a continuous function satisfying an estimate

$$|u(\xi, \eta)| \leq I_q(\xi, \eta) \exp[\eta \cdot I_q(\xi, \eta)]. \tag{17}$$

It also follows from (17) that series (11) is convergent for summable potentials that may be approximated by continuous potentials.

Going back to functions $K$ and $P$, we derive inequalities

$$|K(x, t)| \leq I_q\left(\frac{t+x}{2}, \frac{t-x}{2}\right) \exp\left[\left(\frac{t-x}{2}\right) I_q\left(\frac{t+x}{2}, \frac{t-x}{2}\right)\right],$$

$$|P(x, t)| \leq \left(\frac{t}{x}\right)^\alpha I_q\left(\frac{t+x}{2}, \frac{t-x}{2}\right) \exp\left[\left(\frac{t-x}{2}\right) I_q\left(\frac{t+x}{2}, \frac{t-x}{2}\right)\right].$$

Let us transform the value $I_q$ to estimates

$$I_q\left(\frac{t+x}{2}, \frac{t-x}{2}\right) = \frac{1}{2} \int_{\frac{t+x}{2}}^{\infty} P_{\alpha-1}\left(\frac{y^2(t^2+x^2) - (t^2-x^2)}{2txy^2}\right) |p(y)| \, dy.$$

Therefore, we derive the desired estimate as in Theorem 1.

To finish the proof of Theorem 1, we must to justify the existence of the second continuous derivatives of the function $P(x, t)$ in variables $x, t$ under the condition $q \in C^1(x > 0)$. Obviously, this is equivalent to the existence of the second continuous derivatives of the function $u(\xi, \eta)$ in variables $\xi, \eta$. This is proven by exactly the same method of iterations as above.

Additionally, therefore, Theorem 1 is completely proven.

Now, let us list some classes of potentials that are covered by condition (9). If $|q(x)|$ is monotone decreasing, then it suffices to take $p(x) = |q(x)|$. For potentials with an arbitrary singularity at zero and increasing for $0 < x < M$, say Coulomb ones $q = -\frac{1}{x}$, and which are cut by zero at infinity , i.e., $q(x) = 0$, $x > M$, it suffices to take $p(x) = |q(M)|$ and $x < M$, $p(x) = 0$, $x \geq M$. In addition, condition (9) will be valid for potentials obeying an estimate $q(x + \tau) \leq c|q(x)| = |p(x)|$ (this remark belongs to V.V. Katrakhov).

In particular, the next potentials are covered by condition (9) and are important in applications: a strongly singular potential with power singularity $q(x) = x^{-2-\varepsilon}$, different Bargmann potentials

$$q_1(x) = -\frac{e^{-ax}}{(1 + \beta e^{-ax})^2}, \quad q_2(x) = \frac{c_2}{(1 + c_3 x)^2}, \quad q_3(x) = \frac{c_4}{ch^2(c_5 x)},$$

and Yukawa potentials

$$q_4(x) = -\frac{e^{-ax}}{x}, \quad q_5(x) = \int_x^{\infty} e^{-at} \, dc(t).$$

(cf. [10]).

Remark. In fact, in the proof of Theorem 1, we do not need an explicit form of the Riemann function (5). Only the existence of the Riemann function, its positivity and some special monotonicity property (15) are used. These facts are rather general, so the results of this paper may be generalized to more classes of differential equations.

The estimate from Theorem (1) for a general class of potentials may transform to be a less precise but more simple one.

**Theorem 2.** *Let conditions of Theorem* (1) *be fulfilled. Then, for the transmutation kernel* $P(x,t)$ *the next estimate is valid*

$$|P(x,t)| \le \frac{1}{2}\left(\frac{t}{x}\right)^{\alpha} P_{\alpha-1}\left(\frac{t^2+x^2}{2tx}\right) \int\limits_x^{\infty} |p(y)|\, dy \times$$

$$\times \exp\left[\frac{1}{2}\left(\frac{t-x}{2}\right) P_{\alpha-1}\left(\frac{t^2+x^2}{2tx}\right) \int\limits_x^{\infty} |p(y)|\, dy\right].$$

Let us note that, at $x \to 0$, the kernel of the integral representation may have exponential singularity.

## 3. Kernel Estimates for Power Singular at Zero Potentials

For a class of potentials with a power singularity of the kind

$$q(x) = x^{-(2\beta+1)}, \ \beta > 0, \tag{18}$$

it is possible to simplify the above received estimates to not lost their sharpness. Conditions on $\beta$ are duly needed for summability to infinity.

**Theorem 3.** *Consider potentials of the form* (18). *In this case, Theorem* 1 *holds true with an estimate*

$$|P(x,t)| \le \left(\frac{t}{x}\right)^{\alpha} \frac{\Gamma(\beta)4^{\beta-1}}{(t^2-x^2)^{\beta}} \times P_{\alpha-1}^{-\beta}\left(\frac{t^2+x^2}{2tx}\right).$$

$$\times \exp\left[\left(\frac{t-x}{x}\right) \frac{\Gamma(\beta)4^{\beta-1}}{(t^2-x^2)^{\beta}} P_{\alpha-1}^{-\beta}\left(\frac{t^2+x^2}{2tx}\right)\right],$$

*where* $P_{\nu}^{\mu}(\cdot)$ *is the Legendre function* [16], *a value* $\beta$ *is defined by potential* (18) *and a value* $\alpha$ *by a parameter in the initial Equation,* (2).

Starting the proof, let us note that the estimate needed is derived by a chain of rather long calculations using the Slater–Marichev theorem [17,18]. This theorem is a tool to find many integrals in terms of hypergeometric functions, after reducing them to some forms of the Mellin convolution.

**Proof.** For a class of potentials (17), we will simplify our main estimate (8) from Theorem 1. For this class of potentials, we will simplify an estimate (8), which is a core of the Theorem 1, without any loss of its sharpness. To achieve this, let us calculate explicitly a value $I_q$ (9) from an estimate (8).

The proof of this theorem will be derived from two lemmas.

**Lemma 4.** *For a class of potentials* (17) *the next holds true*

$$I_q(\xi,\eta) = \frac{1}{4\xi^{2\beta}} \int\limits_0^1 P_{\nu}(2\alpha z + 1)(1-z)^{\beta-1}\, dz \tag{19}$$

*where* $P_{\nu}$ *is the Legendre function,* $\alpha = \eta^2/(\xi^2-\eta^2)$.

**Proof.** Consider a value

$$I_q(\xi,\eta) = \frac{1}{2}\int\limits_{\xi}^{\infty} P_{\nu}\left(\frac{t^2(\xi^2+\eta^2)-2\xi^2\eta^2}{t^2(\xi^2-\eta^2)}\right) \frac{dt}{t^{2\beta+1}}.$$

Change variables denoting an argument of the Legendre function as $x$,

$$x = \frac{t^2(\xi^2 + \eta^2) - 2\xi^2\eta^2}{t^2(\xi^2 - \eta^2)}, \; dx = \frac{4\xi^2\eta^2}{t^3(\xi^2 - \eta^2)}dt.$$

Under this change of variables, the limits of integration turn to

$$1, \; 1 + \frac{2\eta^2}{\xi^2 - \eta^2} = \frac{\xi^2 + \eta^2}{\xi^2 - \eta^2} = B > 1,$$

and for a variable $t$, the following formula holds true

$$t = \xi\eta \left( \frac{2}{\xi^2 + \eta^2 - x(\xi^2 - \eta^2)} \right)^{\frac{1}{2}}.$$

It follows, for $I_q$:

$$I_q(\xi, \eta) = \frac{1}{2} \int_1^B P_\nu(x) \frac{t^3(\xi^2 - \eta^2)}{4\xi^2\eta^2} \frac{dx}{t^{2\beta+1}} =$$

$$= \frac{1}{2} \int_1^B P_\nu(x) \left[ \frac{\xi^2 - \eta^2}{4\xi^2\eta^2} \right] \cdot \left[ \frac{\xi^2 + \eta^2 - x(\xi^2 - \eta^2)}{2\xi^2\eta^2} \right]^{\beta-1} dx.$$

Perform one more change of variables in the last integral

$$z = (x - 1)\frac{\xi^2 - \eta^2}{2\eta^2}, \; dz = \left( \frac{\xi^2 - \eta^2}{2\eta^2} \right) dx.$$

As a result, it follows

$$I_q(\xi, \eta) = \frac{1}{2} \left( \frac{\xi^2 - \eta^2}{4\xi^2\eta^2} \right) \int_0^1 P_\nu(2\alpha z + 1) \frac{2\eta^2}{\xi^2 - \eta^2} \cdot$$

$$\cdot \left[ \frac{\xi^2 + \eta^2 - (\xi^2 - \eta^2)\left( \frac{2\eta^2}{\xi^2 - \eta^2}z + 1 \right)}{2\xi^2\eta^2} \right]^{\beta-1} dz = \frac{1}{4\xi^{2\beta}} \int_0^1 P_\nu(2\alpha z + 1)(1 - z)^{\beta-1} dz,$$

where we denote $\alpha = \eta^2/(\xi^2 - \eta^2)$. Therefore, we derive the formula (19) and prove Lemma 4. □

**Lemma 5.** *Let $a > 0$, $\beta > 0$. Then, the next formula holds:*

$$\int_0^1 P_\nu(2\alpha x + 1)(1 - x)^{\beta-1} dx = \Gamma(\beta) \left[ \frac{1 + \alpha}{\alpha} \right]^{\frac{\beta}{2}} P_\nu^{-\beta}(2\alpha + 1). \tag{20}$$

**Proof.** For the proof, we use a technique based on the Slater–Marichev theorem, cf. [17,18].
Change variables in the integral (20) $t = 1/x$, so $x = 1/t$, $dx = \left( -\frac{1}{t^2} \right) dt$. It follows

$$\int_0^1 P_\nu(2\alpha x + 1)(1 - x)^{\beta-1} dx = \int_\infty^1 P_\nu\left( 2\frac{\alpha}{t} + 1 \right)(1 - 1/t)^{\beta-1}\left( -\frac{1}{t^2} \right) dt =$$

$$= \int_1^\infty P_\nu\left(2\frac{\alpha}{t}+1\right)(t-1)^{\beta-1}t^{-\beta}\frac{dt}{t} = \int_0^\infty P_\nu\left(2\frac{\alpha}{t}+1\right)(t-1)_+^{\beta-1}t^{-\beta}\frac{dt}{t} = I(\alpha).$$

Here, we use a notion for the cut power function as $x_+^\lambda$, defined by

$$x_+^\lambda = \begin{cases} x^\lambda, & \text{for } x \geq 0, \\ 0, & \text{for } x < 0, \end{cases}$$

and prolong the integral to the segment $[0,1]$ in $t$, as the cut power function $(t-1)_+^{\beta-1}$ equals zero on this segment. Apply to function $I(\alpha)$ the Mellin transform in a variable $\alpha$ ($\alpha > 0$). Using the Mellin convolution theorem, we derive ([17,18])

$$M[I(\alpha)](s) = M[P_\nu(2x+1)](s) \cdot M[x^{-\beta}(x-1)_+^{\beta-1}](s).$$

Using, one by one, formulas 6(1), (4), 2(4) from [17], we derive

$$M[I(\alpha)](s) = -\frac{\sin \pi\nu}{\pi}\frac{\Gamma(s)\Gamma(-\nu-s)\Gamma(1+\nu-s)\Gamma(\beta)\Gamma(1-s)}{\Gamma(1-s)\Gamma(1+\beta-s)} =$$

$$= -\frac{\sin \pi\nu}{\pi}\Gamma(\beta)\,\Gamma\begin{bmatrix} s, & -\nu-s, & 1+\nu-s \\ & 1+\beta-s & \end{bmatrix},$$

where we use the Slater's notation for a fraction of gamma-function multiplications. In terms of the Slater–Marichev theorem, it follows

$$(a) = (0), \ (b) = (-\nu, 1+\nu), \ (c) = \varnothing, \ (d) = (1+\beta),$$

$$A = 1, \ B = 2, \ C = 0, \ D = 1.$$

Applying the Slater–Marichev theore,m we derive an expression for the $I(\alpha)$ under conditions $0 < \alpha < 1$:

$$I(\alpha) = -\frac{\sin \pi\nu}{\pi}\frac{\Gamma(1+\nu)\Gamma(-\nu)}{\Gamma(1+\beta)}{}_2F_1(-\nu, 1+\nu; 1+\beta; -\alpha) =$$

$$= \Gamma(\beta)\alpha^{-\frac{\beta}{2}}(1+\alpha)^{\frac{\beta}{2}}P_\nu^{-\beta}(1+2\alpha), \qquad (21)$$

with the use of formula (3) from [16], p. 126, and the gamma function identity, cf. [16]

$$\Gamma(-\nu) = \frac{\pi}{\nu\,\Gamma(\nu)\sin \pi\nu}.$$

For $\alpha \geq 1$ we get another expression formally not the same:

$$I(\alpha) = -\frac{\sin \pi\nu}{\pi}\Gamma(\beta)\times$$

$$\times\left\{\alpha^\nu\,\Gamma\begin{bmatrix} 1+\nu+\nu, & -\nu \\ 1+\beta+\nu & \end{bmatrix}{}_2F_1(-\nu, 1-1-\beta-\nu; 1-1-\nu-\nu; -\frac{1}{\alpha})+\right.$$

$$\left.+\alpha^{-1-\nu}\Gamma\begin{bmatrix} -\nu-1-\nu, & 1+\nu \\ 1+\beta-1-\nu & \end{bmatrix}{}_2F_1(1+\nu, 1-1-\beta+1+\nu; 1+\nu; -\frac{1}{\alpha})\right\} =$$

$$= -\frac{\sin \pi\nu}{\pi}\Gamma(\beta)\cdot\left\{\alpha^\nu\frac{\Gamma(2\nu+1)\Gamma(-\nu)}{\Gamma(1+\beta+\nu)}{}_2F_1(-\nu, -\beta-\nu; -2\nu; -\frac{1}{\alpha})+\right.$$

$$\left.+\alpha^{-1-\nu}\frac{\Gamma(-1-2\nu)\Gamma(1+\nu)}{\Gamma(\beta-\nu)}{}_2F_1(1+\nu, 1+\nu-\beta; 1+\nu; -\frac{1}{\alpha})\right\}.$$

However, from [16], p.131, formula (19), it follows that two forms for $I(\alpha)$, $0 < \alpha < 1$ and $\alpha \geq 1$, coincide. This proves Lemma 5. □

As a consequence, we receive the desired estimate for Theorem 3, it is completely proven. □

The estimate using values like (19) for a singular potential $q(x) = cx^{-2}$ for which $\beta = \frac{1}{2}$. Using [16] in this case, the Legendre function $P_\nu^{-\frac{1}{2}}(z)$ is expressed via elementary functions. It follows that an estimate of Theorem 3 also may be written via elementary functions.

Another case for which a kernel estimate may be further simplified and expressed via elementary functions is the potential (18) $q(x) = x^{-(2\beta+1)}$ with a parameter connection property $\beta = \alpha - 1$.

**Corollary 1.** *Let a parameter connection property be $\beta = \alpha - 1$. Then an estimate of Theorem 3 is reduced to*

$$|P(x,t)| \leq \left(\frac{t}{x}\right)^{\beta+1} \frac{2^{\beta-2}}{\beta}\left[\frac{t^2+x^2}{2tx}\right]^{\beta} \cdot \exp\left[\left(\frac{t-x}{2}\right)\frac{2^{\beta-2}}{\beta}\left[\frac{t^2+x^2}{2tr}\right]^{\beta}\right] =$$

$$= \frac{1}{4\beta}\frac{1}{x^{2\beta+1}}(t^2+x^2)^{\beta}\exp\left[\frac{2^{\beta-2}}{\beta}\left(\frac{t-x}{2}\right)\left(\frac{t^2+x^2}{2tx}\right)^{\beta}\right]. \tag{22}$$

**Proof.** In this case, let us transform an estimate of Theorem 3 to the following:

$$\frac{\Gamma(\beta)4^{\beta-1}}{(t^2-x^2)^{\beta}}P_\beta^{-\beta}\left(\frac{t^2+x^2}{2tx}\right) = \frac{\Gamma(\beta)4^{\beta-1}}{(t^2-x^2)^{\beta}}\frac{2^{-\beta}}{\Gamma(\beta+1)}\left[\left(\frac{t^2+x^2}{2tx}\right)^2 - 1\right]^{\frac{\beta}{2}} =$$

$$= \frac{2^{\beta-2}}{\beta}\frac{1}{(t^2-x^2)^{\beta}}\frac{(t^2-x^2)^{\beta}(t^2+x^2)^{\beta}}{(2tx)^{\beta}} = \frac{2^{\beta-2}}{\beta}\left[\frac{t^2+x^2}{2tx}\right]^{\beta}, \tag{23}$$

with the use of the formula from [16],

$$P_\nu^{-\nu}(z) = \frac{2^{-\nu}}{\Gamma(\nu+1)}(z^2-1)^{\frac{\nu}{2}}, \ z > 1.$$

\>From the above, an inequality for kernel function for $\beta = \alpha - 1$ follows in the form of (22). Therefore, the corollary is proven. □

Let us note that, for $\alpha = 0$ in the above proven formulas, our estimates in Theorem 1 reduce to well known estimates for transmutation kernels for Sturm–Liouville equations.

The technique of this paper is also completely applicable to study of non-classical generalized translations. This problem essentially reduced to connection formulas for solutions to the equation

$$B_{\alpha,x}u(x,y) - q(x)u(x,y) = B_{\beta,y}u(x,y) \tag{24}$$

via solutions of an unperturbed Euler–Poisson–Darboux equation with Bessel operators by all variables. Such connection formulas are direct consequences of transmutation theory [1–4]. For generalized translation operators, cf. [19–21].

## 4. Conclusions

This paper is concerned with a study of a special integral equation. This integral equation arises in many applied problems, including transmutation theory, inverse scattering problems, and the solution of singular Sturm–Liouville and Shrödinger equations. A special integral equation is derived and formulated using the Riemann function of a singular

hyperbolic equation. In the paper, the existence of a unique solution to this equation is proven by the method of successive approximations. The results are applied, for example, to the representation of solutions to Sturm–Liouville equations with singular potentials, such as Bargmann and Miura potentials and similiar ones. The treatment of problems with such potentials are very important in mathematical physics, and inverse, scattering and related problems. The estimates received do not contain any undefined constants, and for transmutation kernels all estimates are explicitly written.

### 5. Designation List

$B_\alpha$ is the Bessel differential operator, Formula (2);
$J_\alpha(x)$ is the Bessel function, Formula (3);
$P(x,t), K(x,t)$—kernels of transmutation operators, Formulas (3) and (4);
$q(x)$ is the potential function, Formula (3);
$\mathbf{P}_\alpha$ is the Poisson transmutation operator, Formula (3);
$\mathbf{S}_\alpha$ is the Sonine transmutation operator, Formula (3);
$R_\nu$ is the Riemann function, Formula (5);
$P_\nu^\mu(\cdot)$ is the Legendre function, Theorem 3.

**Author Contributions:** Conceptualization and methodology S.S.; investigation, writing—original draft preparation, writing—review and editing S.S., A.A.A. Both authors have read and agreed to the published version of the manuscript.

**Funding:** This research received no external funding.

**Data Availability Statement:** Not applicable.

**Conflicts of Interest:** The authors declare no conflict of interest.

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
