# Peer review of "On an Integral Equation with the Riemann Function Kernel"

_axioms, doi:10.3390/axioms11040166_

Round 1
Reviewer 1 Report
The authors investigate the singular Sturm-Liouville and Shrodinger equations.
The special integral equation is derived based on the Riemann function of a singular hyperbolic equation.
It is my belief that the manuscript is worth published in the journal.
However, few minor typos are found
- The end of the proof of the lemma2.3, It is written that "proves the lemma2.2" but it must be lemma2.3.
- The first line of the first equation in the page 8 is not correct though the second line is correct.
- The ordering of the curly bracket and dx in the second equation in the page 8 is not correct.
Adding to that, for the equation after eq.(19), the calculation from the first line to the second line is unclear, so the detailed calculation should be added.
I think that after all the above issues are fixed the paper should be positively considered for the publication.
Author Response
Thank you for the review.
Q:The end of the proof of the lemma2.3, It is written that "proves the lemma2.2" but it must be lemma2.3.
A: corrected
Q:The first line of the first equation in the page 8 is not correct though the second line is correct.
A: corrected
The ordering of the curly bracket and dx in the second equation in the page 8 is not correct.
A: corrected
Adding to that, for the equation after eq.(19), the calculation from the first line to the second line is unclear, so the detailed calculation should be added.
A: corrected, one more formular and definition is added for better understanding, with more explanations in text before and after them.
Reviewer 2 Report
The topic of this paper is interesting and has a novelty.
The paper is well arranged and a great effort has been spent to prepare this paper
In view of the given information I would like strongly to offer the acceptance of this paper in "Axioms".
Author Response
Thank you for the review.
Reviewer 3 Report
see the attached file

Author Response
Thank you for the review.
Q: Since this journal is not specialized for transmutations, before the equation (1) is necessary to give an exact
definition of transmutation as for example in R. W. Carroll, Transmutation and operator differential equations, p. 69.
Then must be explain more clearly which are the operators in this case without writing two equality in the same line (as for example in (1) and (2)).
A: much more detailed information is added to introduction on definition and applications of transmutations.
Q: As minor remarks must replace “theorem 2.1” with “Theorem 2.1” in pp. 3, 5…, “lemma 2.2” with “Lemma 2.2”
in p. 7 and so on.
A: corrected for the whole text.
Q: Also I didn’t found all the papers from References quoted in the article (e.g. [18], [19]).
A: corrected, these references are dropped as unnecessary ones.
Round 2
Reviewer 3 Report
Now, I have not important remarks. However in p. 7 remained two times 'theorem 2.1' instead 'of Theorem 2.1' and I did not found references [22] and [23] quoted in the text.